# Cortistatin as a Novel Multimodal Therapy for the Treatment of Parkinson’s Disease

**DOI:** 10.3390/ijms25020694

**Published:** 2024-01-05

**Authors:** Ignacio Serrano-Martínez, Marta Pedreño, Julia Castillo-González, Viviane Ferraz-de-Paula, Pablo Vargas-Rodríguez, Irene Forte-Lago, Marta Caro, Jenny Campos-Salinas, Javier Villadiego, Pablo Peñalver, Juan Carlos Morales, Mario Delgado, Elena González-Rey

**Affiliations:** 1Department of Cell Biology and Immunology, Institute of Parasitology and Biomedicine Lopez-Neyra (IPBLN), CSIC, PT Salud, 18016 Granada, Spain; ignacioserrano23@ipb.csic.es (I.S.-M.); marta.pedreno@outlook.es (M.P.); juliacastillo@ipb.csic.es (J.C.-G.); ferraz.vi@gmail.com (V.F.-d.-P.); pablo.vargas@ipb.csic.es (P.V.-R.); irene.forte@ipb.csic.es (I.F.-L.); martacm@ipb.csic.es (M.C.); jennycampos@ipb.csic.es (J.C.-S.); mdelgado@ipb.csic.es (M.D.); 2Institute of Biomedicine of Seville (IBiS), Hospital Universitario Virgen del Rocío, CSIC, Universidad de Sevilla, 41013 Sevilla, Spain; fvilladiego@us.es; 3Department of Medical Physiology and Biophysics, Faculty of Medicine, University of Seville, 41009 Sevilla, Spain; 4Centro de Investigación Biomédica en Red sobre Enfermedades Neurodegenerativas (CIBERNED), 28029 Madrid, Spain; 5Department of Biochemistry and Molecular Pharmacology, Institute of Parasitology and Biomedicine Lopez-Neyra (IPBLN), CSIC, PT Salud, 18016 Granada, Spain; pablo@ipb.csic.es (P.P.); jcmorales@ipb.csic.es (J.C.M.)

**Keywords:** cortistatin, neurodegenerative diseases, neuroinflammation, Parkinson’s disease, gliosis

## Abstract

Parkinson’s disease (PD) is a complex disorder characterized by the impairment of the dopaminergic nigrostriatal system. PD has duplicated its global burden in the last few years, becoming the leading neurological disability worldwide. Therefore, there is an urgent need to develop innovative approaches that target multifactorial underlying causes to potentially prevent or limit disease progression. Accumulating evidence suggests that neuroinflammatory responses may play a pivotal role in the neurodegenerative processes that occur during the development of PD. Cortistatin is a neuropeptide that has shown potent anti-inflammatory and immunoregulatory effects in preclinical models of autoimmune and neuroinflammatory disorders. The goal of this study was to explore the therapeutic potential of cortistatin in a well-established preclinical mouse model of PD induced by acute exposure to the neurotoxin 1-methil-4-phenyl1-1,2,3,6-tetrahydropyridine (MPTP). We observed that treatment with cortistatin mitigated the MPTP-induced loss of dopaminergic neurons in the substantia nigra and their connections to the striatum. Consequently, cortistatin administration improved the locomotor activity of animals intoxicated with MPTP. In addition, cortistatin diminished the presence and activation of glial cells in the affected brain regions of MPTP-treated mice, reduced the production of immune mediators, and promoted the expression of neurotrophic factors in the striatum. In an in vitro model of PD, treatment with cortistatin also demonstrated a reduction in the cell death of dopaminergic neurons that were exposed to the neurotoxin. Taken together, these findings suggest that cortistatin could emerge as a promising new therapeutic agent that combines anti-inflammatory and neuroprotective properties to regulate the progression of PD at multiple levels.

## 1. Introduction

During the last decades, Parkinson’s disease (PD) has experienced an unprecedented rise in prevalence, disability, and mortality. PD is the fastest-growing neurological disorder in general and the second most common neurodegenerative condition among adults older than 65 years, affecting more than six million people worldwide [1]. The increase in the global burden of PD could be attributed to diverse factors, including increased life expectancy, environmental influences, and the lack of precise disease-targeted interventions.

PD is a chronic disease characterized by the progressive degeneration of dopaminergic neurons that project from the substantia nigra pars compacta (SNpc) to the striatum [2]. The course of the disease includes motor symptoms, such as rigidity and bradykinesia, as well as non-motor features, including sleep disturbances, anosmia, gastrointestinal dysfunction, and depression [2,3,4]. Although the exact cause of PD remains largely unknown, various pathological mechanisms have been proposed, including protein misfolding and aggregation, mitochondrial dysfunction, and excessive oxidative stress. In the last few years, it has also been accepted that the immune system and the activation of resident inflammatory cells may have a potential role in PD. The contribution of neuroinflammation to the progression of PD is supported by evidence demonstrating an increase in inflammatory mediators, activated inflammatory pathways, and oxidation-induced damage to proteins in the serum, cerebrospinal fluid, and brains of patients with PD, as well as in experimental PD models [3,4,5,6,7,8]. Post-mortem analysis of the SNpc of patients has shown the presence of reactive microglia that can produce different inflammatory toxic mediators and contribute to oxidative stress [5,6,7]. Moreover, astroglial cells have been detected in the injured brain regions of PD patients, although the precise role of different astrocyte populations in the disease process remains uncertain [3,5,6]. Furthermore, a mechanistic link between protein aggregation and enhanced neuroinflammation-driven neurodegeneration has been suggested [9,10,11]. Notably, peripheral immune cells are selectively found in the affected brain areas of PD patients, supporting the involvement of systemic immunity in the neuroinflammatory response [5,6,7]. Accordingly, epidemiological studies have revealed that the use of nonsteroidal anti-inflammatory drugs may reduce the risk of developing PD [12,13,14]. Different compounds with anti-inflammatory properties have shown promise in preserving nigral dopaminergic neurons exposed to various neurotoxic insults in vitro and in animal models. Nevertheless, anti-inflammatory therapy in humans has yielded mixed results, likely because of the complexity of the disease in humans and the intricate molecular interactions of inflammatory pathways [14].

On the other hand, multiple in vitro and animal models of PD indicate that most trophic factors protect dopaminergic neurons from neurotoxins, although only a selected number of them can restore the function of the damaged dopaminergic system [15,16,17]. The limited delivery of trophic factors to the brain, which often fails to reach the appropriate brain region or cross the blood–brain barrier (BBB), has hindered the translation of these promising therapies from preclinical models to human trials.

Considering these unsuccessful therapeutic approaches, it becomes evident that this multifaceted disorder necessitates interventions that encompass the protection of the nigrostriatal pathway from inflammation-induced damage and the stimulation of the endogenous production of neurotrophic factors to safeguard and potentially restore the function of dopaminergic neurons. Consequently, it has been proposed that the ideal therapeutic agent for PD should possess both anti-inflammatory and neuroprotective properties.

Previously, we and others have characterized the immunomodulatory effects of cortistatin, a cyclic neuropeptide that is produced by brain cortical neurons and immune cells [18,19]. Beyond its role in autoimmune conditions [19], cortistatin has demonstrated beneficial activities in cell-based systems and pre-clinical models of brain disorders with neuroimmune interactions such as ischemia, glutamate-induced excitotoxicity, bacterial encephalitis, and neuropathic pain [20,21,22,23,24,25]. Recently, we also found that cortistatin reduced the clinical severity and incidence of experimental autoimmune encephalomyelitis, a model of multiple sclerosis [26], in which the systemic administration of cortistatin led to a significant reduction in the presence of inflammatory infiltrates in the spinal cord, and subsequently decreased demyelination and axonal damage. This therapeutic effect of cortistatin was mediated by affecting both the autoimmune and inflammatory components of the disease, regulating glial activity, and promoting an active program of neuroprotection and regeneration [26]. Interestingly, a deficiency in cortistatin was associated with exacerbated inflammatory responses and glial cell overactivity [26,27]. Collectively, these results suggest a crucial role for cortistatin in modulating both peripheral and central immune responses that affect neurological disorders. The preceding evidence prompted us to investigate the potential effects of cortistatin on the regulation of neuroinflammatory and neurodegenerative processes that occur during the development and progression of PD. In this study, we used the most widely validated preclinical model of parkinsonism, which is induced by the systemic exposition of mice to the toxin 1-methil-4-phenyl1-1,2,3,6-tetrahydropyridine (MPTP) [28]. The MPTP model replicates several key pathological features of the idiopathic form of PD because it causes profound neurodegeneration of the nigrostriatal pathway, along with neuroinflammation and oxidative damage [28]. We evaluated various hallmarks of PD pathology, such as behavior and motor deficits, dopaminergic neuronal function, glial activation, and levels of neurotrophic factors. We also conducted cell culture experiments to identify the cellular targets of the actions of cortistatin.

## 2. Results

### 2.1. Treatment with Cortistatin Reduces MPTP-Induced Locomotor Dysfunction

PD courses include motor deficits such as bradykinesia (slowness of movement), rigidity, postural instability, and rest tremor [29]. Several of these alterations are replicated in the MPTP model and can be measured using standardized tests that assess rodent locomotion and motor coordination [29,30,31]. Therefore, we performed some of these behavioral assays to evaluate the brain functions that are primarily affected by changes in the dopaminergic system seven days after the last MPTP injection (Figure 1a). 

First, in the rotarod test, which measures movement coordination, we observed that mice injected with MPTP exhibited a significantly lower walking time on the rotarod than saline-treated control animals, even after training, whereas treatment with cortistatin significantly increased this retention time (Figure 1b). Similarly, in the pole test, which determines locomotor activity, the descending time from the pole was notably delayed in the MPTP-intoxicated mice compared with the control group and was restored after cortistatin treatment (Figure 1c). Furthermore, we evaluated neuromuscular strength and coordination using the hang and gait tests, respectively. In the hang test, although mice treated with cortistatin did not stay longer on the grid than MPTP-treated mice (data not shown), they exhibited improved hanging performance, characterized by less time of inactivity and significantly increased total running distances and speeds (Figure 1d). Additionally, we observed a reduction in the stride length of both forelimbs and hindlimbs in MPTP-exposed mice, while cortistatin-treated mice showed a higher percentage of individuals with long strides for both footprints, exhibiting a pattern similar to that observed in control mice (Figure 1e). These findings indicate that cortistatin enhances the locomotor activity of MPTP-exposed mice, suggesting a protective effect against neurotoxin-induced dopaminergic denervation.

### 2.2. Cortistatin Attenuates MPTP-Induced Dopaminergic Neurodegeneration

Next, we investigated the impact of this neuropeptide on the integrity of the nigrostriatal pathway by measuring the protein levels of tyrosine hydroxylase (TH) in the mesencephalic and striatal tissue samples. TH is the enzyme responsible for catalyzing the conversion of the amino acid L-tyrosine to L-3,4-dihydroxyphenylalanine (L-DOPA), the precursor for dopamine production [32]. In addition, we monitored temporal changes in the gene expression of *Th* and the dopamine transporter (*Dat*) in both MPTP-affected brain regions and measured striatal dopamine levels. One week after MPTP administration, we observed a striking degeneration of dopaminergic TH^+^ neurons in the SNpc (Figure 2a and Appendix A) along with a reduction in dopaminergic striatal innervation (Figure 2b and Appendix A) compared with control mice treated with saline. This finding was consistent with a substantial and enduring downregulation in the gene expression of both *Th* and *Dat* (Figure 2c) and a significant decline in dopamine levels (Figure 2d). However, systemic administration of cortistatin significantly preserved the number of TH-immunoreactive neurons and nerve fibers in both the SNpc and striatum (Figure 2a–c and Appendix A) and the levels of *Th* expression (at early time points after MPTP administration). 

Remarkably, cortistatin treatment also prevented the dramatic loss of *Dat* in the SNpc caused by MPTP intoxication and restored its homeostatic expression levels in the striatum (Figure 2c). Consequently, the neuropeptide appeared to avoid MPTP-induced progressive loss of striatal dopamine (Figure 2d). Subsequently, considering the safeguarding influence that trophic factors confer upon dopaminergic neurons against neurotoxin-induced cell death, we assessed the impact of cortistatin on brain-derived neurotrophic factor (BDNF), which is recognized as a principal factor that is involved in maintaining the structural integrity of neurons in animal models of PD [15,16]. Although *Bdnf* expression was not affected by MPTP at early time points, it was significantly reduced in the striatum 7 days after intoxication (Appendix A). Interestingly, the striatum of cortistatin-treated mice revealed similar *Bdnf* levels to those found in saline-injected control mice at this time point (Appendix A). 

These results collectively suggest that cortistatin exerts a beneficial effect on the MPTP-induced acute PD model by preserving the functionality of the cellular and molecular dopaminergic system.

### 2.3. Cortistatin Modulates the Neuroinflammatory Response Induced by Acute MPTP Administration

Although the precise molecular mechanisms responsible for MPTP-induced dopaminergic neurotoxicity remain unclear, glial dysfunction may be a significant contributor to the pathogenesis of human PD and the MPTP-induced PD model [6,33,34,35]. Consequently, our study aimed to explore the impact of cortistatin on MPTP-induced glial activation in the nigrostriatal pathway by assessing glial density and reactivity by measuring the expression of the glial fibrillary acidic protein (GFAP) for astrocyte detection and the expression of the ionized calcium-binding adaptor molecule 1 (Iba1) for quantifying reactive microglia. One week after MPTP injection, the SNpc and striatum of control saline-injected mice exhibited minimal detection of GFAP^+^ astrocytes and Iba1^+^ microglia. However, a robust astrocytic response and increased reactive microglia were observed after one week of MPTP intoxication in both brain regions, as evidenced by the extensive area covered by GFAP^+^ and Iba1^+^ cells, along with elevated expression of both glial markers (Figure 3a–c, Figure 4a–c, Appendix A).

Systemic administration of cortistatin significantly attenuated MPTP-triggered astrogliosis and microgliosis in the SNpc (Figure 3a,c, Figure 4a,c, Appendix A). Conversely, cortistatin treatment had no discernible effect on astrocytic and microglial density and activation in the striatum of MPTP-treated mice (Figure 3b,c, Figure 4b,c, Appendix A).

Notably, microscopic analysis revealed distinct morphological characteristics between microglia found in the nigrostriatal system of the saline-treated, MPTP-intoxicated, and cortistatin-treated animals (high magnification images from Figure 4a,b and Figure 5a). Although notable differences were primarily observed in the SNpc, they were also evident in the striatum (Figure 5a). Because microglia are known to adopt different phenotypes during the progression of neurodegenerative injuries [36,37,38,39], we further characterized the modulatory effect of cortistatin on glial activation. Thus, we evaluated microglial characteristics such as shape and size (Figure 5b), ramification (Figure 5c), and complexity (Figure 5d), all of which correlate with different grades of cell activation. We found that MPTP administration resulted in larger microglia cell bodies, an increased area and perimeter, and more branches and cell junctions in both SNpc and striatum compared with those observed in control mice, which displayed a physiological morphology characterized by small cell bodies with long thin processes arising from the cytosol, representing mildly ramified cells (Figure 5b,c and Appendix A). Notably, microglia in the SNpc exhibited thicker and shorter projections than those observed in the striatum of MPTP-intoxicated mice (Figure 5c and Appendix A). Fractal analysis also indicated increased complexity in the microglia from the SNpc of MPTP-treated mice, which was accompanied by an ameboid shape (evaluated using circularity and span ratio shape descriptors, Figure 5d and Appendix A). Interestingly, microglia from cortistatin-treated mice displayed a mixed phenotype resembling those in control mice. Cortistatin significantly reduced microglia size in both SNpc and striatum and modulated the network of ramifications to exhibit fewer but longer branches than those of microglia from MPTP-treated mice (Figure 5a–c and Appendix A). Furthermore, cortistatin administration reversed MPTP-induced reactive microglia to a physiological morphotype print (Figure 5d and Appendix A).

Next, we decided to focus on examining the temporal dynamics of genes linked to glial activation, namely, *TNF-α* and *IL-1β*, within the context of the acute MPTP model with or without cortistatin intervention (Figure 6). Our observations revealed an upregulation of these pro-inflammatory cytokines two days after MPTP administration in both SNpc and striatum, which was sustained for up to seven days (Figure 6). The treatment with cortistatin significantly prevented the exacerbated expression of both pro-inflammatory factors during the progression of the MPTP model in both brain regions (Figure 6). 

Together, these findings suggest that cortistatin exerts a modulatory effect on MPTP-induced gliosis, particularly in the SNpc. Cortistatin not only reduces astrocyte/microglia density and their reactive response but also regulates the morphological dynamics of microglia in an injury-induced environment, leading to microglial morphotypes that are comparable to those described for healthy and physiological microglia.

### 2.4. Cortistatin Alleviates the Peripheral Immune Response Induced by MPTP Intoxication 

In addition to the chronic neuroinflammation associated with PD pathophysiology, notably elevated levels of cytokines and T cells have been identified in both the serum and cerebrospinal fluid of patients with PD, providing evidence for the potential interaction between systemic inflammation and neurodegenerative processes in PD [5,6,7,40]. Consequently, we decided to assess the impact of cortistatin on the systemic immune response within this PD model. MPTP administration induced early systemic inflammation in mice, as demonstrated by the increased serum levels of proinflammatory cytokines, such as *IL-6* and *TNF-α*, and the chemokine CCL2 (Figure 7) at two days post-intoxication. Treatment with cortistatin markedly attenuated the systemic levels of these mediators, suggesting an immunoregulatory role of cortistatin in mitigating MPTP-induced peripheral inflammation (Figure 7).

### 2.5. Protective Effect of Cortistatin on Dopaminergic Neurons Exposed to MPP^+^

Apart from its neuroprotective influence on the dopaminergic system through modulation of glial activation, cortistatin may directly impact the survival of compromised dopaminergic neurons. To investigate this, we examined the effect of cortistatin on 1-methyl-4-phenylpyridinium (MPP^+^)-induced dopaminergic neurodegeneration. The neurotoxin MPTP is converted by the enzyme MAO-B in the brain into MPP^+^, a functional metabolite that causes toxicity specifically to dopaminergic neurons [41]. We used mixed primary fetal ventral midbrain neuronal-enriched cultures consisting of 70–80% neurons (of which approximately 1–2% are TH^+^), and 10–15% glial cells, primarily astrocytes (Figure 8a). 

As shown in Figure 8a,b, these mesencephalic cultures with MPP^+^ resulted in a significant reduction (nearly 70% of control cultures) in the number of TH^+^ neurons. Of note, MPP^+^ also affects, although to a lesser extent, the viability of non-dopaminergic neurons (determined using the pan-neuronal marker MAP2). Interestingly, the addition of cortistatin to the cultures significantly protects both TH^+^ and MAP2^+^ neurons from MPP+-induced toxicity. We also found that after MPP^+^, more than one-half of the remaining dopaminergic neurons exhibited either complete (type 2 neurons) or partial (type 3 neurons) loss of dendrites (Figure 8a,c). The addition of cortistatin to the cultures significantly mitigated the pronounced shortening or complete disappearance of the TH^+^ cell processes induced by MPP^+^ (Figure 8). This finding suggests that cortistatin may directly shield intoxicated dopaminergic neurons from complete dendritic loss and protect non-dopaminergic cells, thereby indicating a potential role in neuroprotection.

## 3. Discussion

PD is one of the most complex disorders affecting the CNS. It is characterized by the interplay of various factors, including dopamine depletion resulting from the loss of dopaminergic neurons, glial activation, and a reduction in growth factors [2,3,4]. Because of its intricate pathophysiology, there is no effective cure for this disorder. Current therapies primarily focus on maintaining dopamine levels, achieved either by inhibiting its endogenous degradation or by supplementing dopamine through precursors or agonists. However, these treatments are essentially symptomatic, focusing on improving functional mobility and extending the life expectancy of patients with PD. Given the multifactorial pathobiology of PD, it could be highly promising to search for novel agents designed to simultaneously target several aspects. To achieve this, it is crucial to gain a deeper understanding of the pathogenic mechanisms underlying the onset and progression of PD. Despite some controversy in the field, recognizing the role of inflammation in PD could have a positive impact on the development of new therapeutic pathways. 

Cortistatin is a neuropeptide with a critical anti-inflammatory role that participates in brain–immune communication (reviewed in [19]). In this study, we investigated whether cortistatin could regulate the relationship between neuroinflammation and neurodegeneration in an experimental model of PD and exert any beneficial effect. First, our results demonstrated that systemic administration of this neuropeptide conferred protection against Parkinsonian toxicity induced by MPTP in mice. Notably, cortistatin mitigated dopaminergic neuron death in the SNpc and shielded against the loss of dopaminergic nerve terminals in the striatum. The restoration of dopamine levels and improvement in locomotor activities following MPTP challenge in cortistatin-treated mice suggest the essential role of this neuropeptide in preserving a functional nigrostriatal system.

Regarding the mechanism of action, one possibility is that cortistatin may prevent dopaminergic cell death. Thus, given its demonstrated neuroprotective effects in various experimental models [20,21,22,23,24], and its ability to inhibit MPP^+^-induced cell loss while maintaining the healthy morphology of dopaminergic neurons (Figure 8), cortistatin may act directly to avoid neuronal toxicity. In addition, cortistatin may indirectly protect ventral midbrain dopamine neurons by preserving nigrostriatal projections. This is particularly relevant considering reports indicating the presynaptic terminals of dopaminergic neurons as the initial site of injury in acute MPTP models and that the loss of connectivity with the target population may underlie phenotypic abnormalities in the disease [42]. In our study, cortistatin effectively protected against severe depletion of dopaminergic neurons and projections in both the SNpc and striatum of MPTP-intoxicated animals. Furthermore, while striatal TH expression was recovered in both MPTP- and cortistatin-treated mice to levels found in healthy animals one week after intoxication, DAT and BDNF levels were preserved (just in the striatum) only in cortistatin-treated mice. DAT is a sensitive indicator of damage to dopaminergic neurons; therefore, the action of cortistatin in avoiding a sustained reduction in DAT expression could influence the recovery of the nigrostriatal pathway after MPTP intoxication. Regarding BDNF, it is strongly expressed by dopaminergic neurons in the SNpc and it is reduced by 70% in PD patients, partly due to the loss of dopaminergic neurons that express BDNF [43]. Protective effects in animal models of PD have been obtained with BDNF and other trophic agents, including glial cell line-derived neurotrophic factor and basic fibroblast growth factor [44,45,46]. However, their clinical application has been limited due to delivery challenges and side effects. For example, BDNF does not readily diffuse across the BBB or ventricular lining, and it has limited or unstable bioavailability and potential toxicity [45,47]. Hence, the induction of endogenous BDNF is highly desirable. Recently, an intriguing connection has been described between BDNF and cortistatin [48,49]. Beyond the effect of cortistatin on BDNF expression in the acute MPTP injection model, we have reported an in vivo modulation of BDNF levels by cortistatin in a progressive model of MS [26] and neuropathic pain [25], suggesting a broader impact of cortistatin on neurotrophic factors beyond specific disease contexts. 

Second, we observed that the positive effect of cortistatin on MPTP-induced neurodegeneration was accompanied by the modulation of an exacerbated neuroinflammatory response and the concomitant release of glial-derived inflammatory mediators in brain regions affected by MPTP. This immunoregulatory effect of cortistatin in the CNS was extended to the periphery at the systemic level. Multiple lines of evidence suggest the involvement of neuroinflammatory processes in the pathophysiology of PD, although their origin and precise role remain uncertain [2,3,4,5,6,7,8]. In the acute model of murine Parkinsonism, MPTP initiates a self-perpetuating neurodegenerative process wherein glial cells are involved from the onset, reaching peak activity one day post-intoxication and persisting for approximately a week [50]. Upon systemic administration, MPTP crosses the BBB and is metabolized by glial cells to MPP^+^, the active toxic form of the molecule, which is subsequently released into the extracellular space [41]. Because of its high affinity for the DAT transporter, MPP^+^ accumulates in dopaminergic neurons, where it inhibits mitochondrial electron transport chain complex I. This leads to enhanced production of reactive oxygen species, diminished ATP synthesis, and subsequent neuronal distress [41]. This phase lasts for approximately 3–4 days and is followed by the destruction of neuronal cell bodies. One week after exposure to MPTP, dopaminergic cell death did not progress further. During this timeframe, reactive microglia can rapidly respond (12–24 h post-MPTP) even to mild neuropathological changes, whereas astroglial reactions occur later, coinciding with dopaminergic cell death [3]. Numerous animal studies indicate that this glial crosstalk can fuel a cycle of progressive dopaminergic neurodegeneration [50,51,52,53], potentially preceding neuronal death [54,55]. In addition, the inflammatory response of reactive glia contributes to the site-specific recruitment of peripheral immune cells [56,57].

Based on these observations, we demonstrated that MPTP-induced gliosis in both the SNpc and striatum was accompanied by the local and systemic production of inflammatory mediators. This suggests that preventing glial responses may impede neuronal degeneration. However, addressing this challenge is considerably intricate, as evidenced by findings from preclinical models and clinical trials. Microglia exhibit significant spatiotemporal variation, manifesting distinct morphologies and molecular signatures. Recent studies employing various techniques and integrated parameters have revealed a direct correlation between microglial phenotypes and functionality [36,37]. Moreover, signals from surrounding cells may affect microglial morphotypes [58]. Microglial phenotypes in the midbrain differ from those found elsewhere in the CNS, which could be related to the selective vulnerability of dopamine neurons in PD [59]. Far from the classical “resting” versus “activated” dualistic classification, different functional states of microglia, indicative of high dynamics and plasticity, are identified. A deeper characterization of these mixed phenotypes is crucial for understanding health and disease, particularly tissue repair. In this context, our study revealed that the microglial population in the SNpc and striatum of MPTP-exposed animals mainly comprised highly ramified complex cells. However, these cells did not fit the typical surveillance phenotype (type 1, Appendix A) because of their large cell soma and thick, short branches, which resembles a more reactive morphology (type 3, Appendix A). This phenotype of microglia that is induced by MPTP in vivo has been linked to exacerbated phagocytosis of MPP^+^-damaged dopaminergic neurons as well as viable neurons within a neuroinflammatory environment [39,60,61,62], which could potentially cause harm to the surrounding parenchyma. Conversely, Iba1^+^ cells in cortistatin-treated mice displayed a mixed phenotype with bipolar and slightly ramified cells, characterized by small soma and thin processes (type 1 + 2, Appendix A), a morphology similar to that observed in the brain parenchyma of healthy animals. Notably, although no significant differences in microgliosis were observed between untreated and cortistatin-treated MPTP mice, these morphological analyses enabled the distinction of various morphotypes in striatal microglia which are likely associated with different functionality.

Similar to microglia, astrocytes exhibit regional heterogeneity because they share unique functional and morphological properties that may contribute to the selective vulnerability observed in PD. Dysfunctional astrocytes and reactive astrogliosis also play a role in the degeneration of dopaminergic neurons [53,63,64,65,66]. Our results showed a significant reduction in nigral astrogliosis in cortistatin-treated mice; however, no changes were observed in the striatal astrocytes. Interestingly, because astrocytes are the main suppliers of neurotrophic factors and strictly regulate the integrity of the BBB, recent studies have revealed that the loss of a specific astrocytic population (with homeostatic supporting roles) is implicated in the onset and progression of PD [52]. Further analysis will confirm whether cortistatin can differentially modulate reactive and protective astrocytes.

The modulation of local inflammatory mediators produced by both glial populations suggests that cortistatin plays an immunoregulatory role rather than merely a deactivating effect in shaping the neuroinflammatory response. As previously reported, several inflammatory mediators have been implicated in neurodegeneration in both MPTP models and PD patients [67]. Among these, microglia-derived *TNF-α* and *IL-1β* can exert neurotoxic effects by directly activating receptors on dopaminergic neurons, triggering cell death pathways, or through indirect mechanisms via glial activation [68,69,70]. In addition, astrocytic CCL2-mediated monocyte recruitment has been associated with increased dopaminergic neuronal loss [71]. While the dual functions of IL-6 in the CNS include supporting immunoregulatory and neurotrophic functions [72], evidence suggests that downregulating the heightened IL-6 secretion during PD can reduce damage to dopaminergic neurons [67,70,73]. Therefore, the ability of cortistatin to target both brain and peripheral immune activation may break the vicious cycle and modulate the neurodegenerative process, contributing to its neuroprotective activity. Several agents, including other neuropeptides, have been identified as neuroprotective because of their capacity to reduce microglial responses in different PD models [56,74,75,76,77]. Hence, the modulation of glial activation by cortistatin may be crucial in preventing glial-mediated inflammation and neuronal depletion, while promoting trophic and protective functions. Considering that microglial activation is initiated in early PD and remains stable for years [62,78], it is essential to explore treatments that disrupt the continuity of reactive microglia. Furthermore, although we and others have previously highlighted the regulation of microglial functions by cortistatin in neuroinflammatory and neurodegenerative conditions [25,26], this report provides the first detailed account of the fine-tuning of the plastic repertoire of microglial states by cortistatin. 

In addition to neuroinflammation, oxidative stress is widely acknowledged as a crucial pathophysiological event contributing to the progressive loss of nigral dopaminergic neurons in PD [79]. Considering that cortistatin reduces the production of inflammatory cytokines and nitric oxide by reactive glial cells [26], it is plausible that, alongside its anti-inflammatory effects, this neuropeptide could induce neuroprotection in the MPTP model through its antioxidant properties.

Cortistatin exerts its effect through somatostatin receptors (Sstr1–5) and the ghrelin receptor (Ghsr1a). Data from the Human Protein Atlas [80,81] and the Allen Brain Institute [82] indicate that cortistatin and SSTR1–4 are expressed in the midbrain and striatum of both humans and mice, whereas SSTR5 and GHSR1a are barely detectable. However, GHSR1a expression has been reported in MES35.5, a dopaminergic neuronal cell line [83], and dopaminergic neurons of the SNpc [84]. Interestingly, the neuropeptide ghrelin, acting through GHSR1a, has also exerted therapeutic effects in MPTP-intoxicated mice [84]. These findings suggest that the protective effect of cortistatin observed in our study may be mediated through these receptors. The potential capability of cortistatin to synergistically signal through somatostatin/ghrelin receptors (in addition to a still unidentified specific receptor) could offer an advantage over the individual neuroprotective effect exerted by somatostatin/ghrelin. Further research will address the specific contribution of each receptor to the beneficial effect of cortistatin. 

The precise role of endogenous cortistatin in this complex disorder is still under investigation. However, evidence from our laboratory and others suggests that a deficiency in cortistatin predisposes patients and animalsto a more robust local immune response and neurodegeneration [25,26,27,85,86]. Importantly, we recently reported that the absence of endogenous cortistatin leads to dysfunction in the brain endothelium, and treatment with cortistatin restores the properties of the brain endothelial barrier [87]. Preliminary experiments performed in our laboratory indicated that mice deficient in cortistatin have a higher and earlier mortality (50.91 ± 11.65% of mice) than wild-type mice when exposed to MPTP (10.21 ± 2.58% of mice), suggesting a main role of this neuropeptide in protecting against the initiation of the disease. It is conceivable that a compromised BBB in the absence of cortistatin could contribute to increased susceptibility to the neurotoxic effects of MPTP. A previous study by our group demonstrated that the absence of this neuropeptide in nociceptive neurons exacerbated pain responses to inflammatory stimuli [25,88], suggesting that endogenous cortistatin plays an important role in the function of specific neurons. In addition, glial cells from mice lacking cortistatin exhibited an exacerbated immune response in an inflammatory context (unpublished data). Collectively, these findings suggest that endogenous cortistatin likely protects against neurodegenerative and neuroinflammatory pathologies. Further experiments will determine whether the absence of cortistatin predisposes patients to an exacerbated disease and whether cortistatin deficiency affects the nigrostriatal pathway in terms of susceptibility to neuronal death and enhanced neuroinflammation. 

In summary, the role of neuroinflammation in PD is intricate and it may appear either because of neuronal degeneration or as a primary cause of the disease. Regardless of its origin, the development of therapeutic interventions aimed at preventing or downregulating immune-associated mechanisms holds promise for impairing disease progression or even arresting the pathological process. However, PD is a multifactorial disorder with numerous local and systemic cellular and molecular interactions, each playing a dual role throughout the disease. This complexity might explain why therapies targeting a single mechanism have faced challenges, making complementary treatments a more appealing approach. Our findings suggest that cortistatin emerges as a potential novel therapeutic agent with both immunoregulatory and neuroprotective properties and is capable of influencing the progression of experimental parkinsonism at multiple levels. The beneficial effect of cortistatin in modulating glial populations concerning their protective, immune, and trophic functions, as well as its impacts on peripheral inflammation, highlight that cortistatin could have an important role as a therapy, not only for PD but also for other neuroinflammatory and neurodegenerative disorders. However, caution is warranted when translating our findings from the experimental model to clinical practice in patients with PD. This caution arises from the fact that MPTP-induced nigrostriatal neurodegeneration does not fully mimic the pathological mechanisms that cause PD in humans and because some controversy still exists regarding the consideration of neuroinflammation as a real therapeutic target in this multifaceted disorder.

## 4. Materials and Methods

### 4.1. Animals and Ethics Statement

The experiments reported in this study followed the ethical guidelines for investigations of experimental animals approved by the Animal Care and Use Board and the Ethical Committee of the Spanish Council of Scientific Research. They were performed in accordance with the guidelines outlined in Directive 2010/63/EU of the European Parliament concerning the protection of animals used for scientific purposes and were reported to comply with the ARRIVE v.2.0 guidelines [89]. Male mice (20–24 g body weight, 12 weeks old, supplied by Charles River) were used in all experiments. Animals were housed in a controlled-temperature/humidity environment (22 ± 1 °C, 60–70% relative humidity) with a 12 h light/dark cycle (lights on at 7:00 a.m.) and were fed with rodent chow (Global Diet 2018, Harlan, KY, USA) and tap water ad libitum. Mice were randomly assigned to different experimental groups (5–8 mice per group and cage) following the software G*Power (www.gpower.hhu.de, accessed on 14 October 2023; RRID:SCR_013726) to obtain a power > 0.95 to detect a change of approximately 30%, assuming a standard deviation of 30% at a significance level of *p* < 0.05, expecting an effect size of 1.82 for ANOVA tests. 

### 4.2. Reagents

Mouse cortistatin-29 was purchased from Bachem (Bubendorf, Switzerland), and MPTP and MPP^+^ iodide were purchased from Sigma-Aldrich (St. Louis, MO, USA). Cortistatin was dissolved in physiological saline (0.9% NaCl for in vivo studies; 20 mM phosphate buffer pH 7.0 for in vitro assays). MPTP was dissolved in phosphate buffer saline (PBS) with a pH of 7.2 and MPP^+^ was dissolved in ddwater.

### 4.3. Induction and Treatment of the PD Model

To induce PD, male C57BL/6 mice (12 weeks old, Charles River) were injected intraperitoneally with MPTP (20 mg/kg, four times at intervals of 2 h). Treatment consisted of intraperitoneal (i.p.) injection of PBS as a vehicle (MPTP mice) or cortistatin (2 nmol, MPTP + CST) starting 2 h after the last MPTP injection and then once daily for 6 additional days. Naive mice injected with PBS (saline) instead of MPTP were used as sham controls. Mice were euthanized using deep anesthesia (i.p. injection of 350 mg/kg chloral hydrate) 2 and 7 days after the last injection with MPTP or saline and intracardially perfused with PBS and 4% paraformaldehyde (PFA). Brains were isolated and processed for immunohistochemistry and RNA extraction, as described below. Mice sacrificed on day 7 were subjected to locomotor activity testing before sacrifice.

### 4.4. Evaluation of Motor Activity

- Rotarod test: Mouse motor coordination and balance were assessed using a rotarod apparatus (Harvard Apparatus, model LE8205). Before measurements, mice underwent three consecutive training trials with the rod at constant speeds: one trial at 0 rpm and two trials at 4 rpm. The test began when the animals could stay on the rod for 60 s. Each mouse was individually placed for four consecutive trials (20 min inter-trial intervals) on the rod, which uniformly accelerated from 4 to 40 rpm in 300 s. The latency to fall was recorded in seconds in each trial, and the mean for each mouse was calculated as the average of the four trials. 

- Pole test: To evaluate coordination and movement disorders, a wooden pole (diameter: 1 cm; length: 45 cm) with gauze on its surface was positioned in the home cage. After acclimatization, the mice were pre-trained at least three times to ensure that all animals would turn their heads down once placed on the pole. The turning and descending times of each mouse from the top to the bottom of the pole were recorded. The results represent the average values of three trials.

- Gait test: This test measures basal ganglia-related movement disorders. The mice were placed on an illuminated runway (4.5 cm wide, 50 cm long) and allowed to run toward their home cage. Before the test, the animals were acclimatized over two trials. Stride length was measured for the forelimbs and hindlimbs, differentiating them by marking the front and back paws with blue and black ink, respectively. Walking on white paper, the footprints were recorded and analyzed. Stride length was measured as the distance between successive paw prints. The average of the three strides was considered for each animal.

- Hang test: This assay measures the neuromuscular strength of mice. Animals were placed on the wire lid of a standard mouse cage (total size 12 cm^2^ with openings of 0.5 cm^2^) until they grabbed the grid. The lid was then inverted, and the mice were allowed to hang upside down. The vertical distance between the wire lid and the bottom of the cage was 30 cm. Each mouse was tested three times with 90 s per trial, and the time that the mouse stood with at least two limbs on the lid was recorded. Walking distance, speed, and resting time were also evaluated.

### 4.5. Histological Analyses: Immunohistochemistry and Immunofluorescence 

After transcardial perfusion of mice, brains were dissected, postfixed for 24 h with 4% PFA at 4 °C, and then dehydrated in a 30% sucrose solution until they sank. Brains were embedded into OCT Tissue-Tek and cryosectioned to obtain serial coronal sections (30 μm) that included the striatum and SNpc. Sections were stored at 4 °C in cryoprotectant solution for further analysis. 

- For immunohistochemistry staining (TH detection), tissue sections were washed with 0.1 M PB with 0.05% triton X-100 and incubated with 1% H_2_O_2_ for 15 min at room temperature. The sections were then blocked for 30 min and incubated with primary rabbit anti-mouse TH antibody (Santa Cruz, Heidelberg, Germany) for 4 °C overnight. Subsequently, the sections were incubated with biotin-conjugated secondary antibody (Vector Laboratories, Newark, CA, USA) for 1.5 h at room temperature and stained with the VECTASTAIN Elite ABC system (Vector Laboratories) following the manufacturer’s protocol. Briefly, ABC reagent was used for 45 min, sections were washed again, and finally revealed with DAB substrate solution until the desired stain intensity was obtained. The sections were then rinsed with cold 0.1 M PB and mounted in 50% glycerol in PBS with 0.03% sodium azide. Images were acquired using a light-transmitted microscope (Leica DM2000; Leica, Wetzlar, Germany). The sections covered the entire rostrocaudal axis of the SNpc (AP, −2.7 to −3.8 mm from Bregma) and striatum (Bregma +0.98 and +0.02 mm) according to the Franklin and Paxinos mouse brain stereotaxic atlas [90], yielding approximately 30–36 sections. A total of five sections (one of every five) were stained, acquired, and analyzed throughout the entire extent of both the SNpc and striatum. 

- For immunofluorescence staining (glial cells detection), tissue sections were washed and blocked as previously described. Subsequently, the sections were incubated overnight (4 °C) with the primary polyclonal antibodies rabbit anti-mouse Iba1 for microglia (Fujifilm Wako Chemicals, Neuss, Germany) and anti-GFAP for astrocytes (Agilent, Santa Clara, CA, USA), followed by incubation with Alexa Fluor 594-labeled anti-rabbit (Thermo Fisher Scientific, MA, USA) for 1.5 h at room temperature. After several washes, the nuclei were stained with DAPI, and the sections were mounted with Mowiol (Sigma). Images were acquired using a LEICA DMi8 S Platform Life-cell microscope and processed as described below.

### 4.6. Stereology and Densitometry Evaluation

The TH-stained neurons in the SNpc were visualized in both the right and left SNpc and quantified using unbiased stereological analyses with systematic random sampling employing the optical dissector method [91]. The CASTTM system (Visiopharm, Hørsholm, Denmark) was used following a previously described methodology [74,75]. In brief, five sections (30 µm thickness/section) at five-section intervals between Bregma −2.92 mm and −3.40 mm were selected. Only SNpc cells lateral to the medial terminal nucleus of the accessory optic tract were considered to have a clear separation from the adjacent ventral tegmental area. The SNpc area for each section was outlined under low magnification (4×), and TH^+^ cell bodies were counted under a 40× objective using a 7225 μm^2^ × 20 μm optical dissector with a guard volume of 5 μm to avoid artifacts on the cut surface of the sections. The results are indicated as the total number of TH-positive neurons. 

The optical density of SNpc and striatal TH^+^ innervation, Iba1^+^, and GFAP^+^ staining (to estimate gliosis) were determined in digitized images using the NIH ImageJ Fiji v2.1.0/1.53c free software (https://Fiji.sc, accessed on 14 October 2023) as previously described [92,93]. Values for each animal were obtained from a total of 3–5 sections (covering the SNpc and striatal regions, respectively), and the percentage of the positive area for each marker was calculated from the total area in a selected region of interest (ROI). For striatal TH innervation, a calibrated optical density step tablet served as a reference for gray values. The concentration of the anti-TH antibody and 3,3′-diaminobenzidine (DAB) and the duration of incubation of sections in DAB were optimized to fall within the linear range of the immunostaining intensities plot. To control for variations in background illumination, the average background density readings from the cortex were subtracted from those in the striatum for each section. 

### 4.7. Microglia Phenotype Characterization

To identify the phenotypic diversity of microglia cells, we examined various morphological parameters in Iba1^+^-immunostained sections from SNpc and striatum using established protocols [36,38,94]. Regions of interest were randomly selected to avoid cell overlap and exclude incomplete cells. We analyzed 65–90 cells from each experimental group, comprising five mice. In brief, images were first processed to reduce background noise and enhance contrast. Subsequently, individual cells were cropped from binary images and morphometric analysis was conducted. Following the manual determination of cell body size, the Skeleton plugin in ImageJ [95,96] was used to evaluate the number of branches, maximum branch length, and total number of junctions. To characterize complex cellular patterns, the FracLac plugin was employed [97]. This involved evaluating the fractal dimension (a statistical index indicating complexity with values ranging from simple rounded shapes to intricate branched structures with higher values); convex hull area (the area containing the entire cell shape enclosed by the smallest convex polygon); perimeter (total length of the boundary of the cell shape); circularity (calculated as (4π × cell area)/(cell perimeter)^2^, with 1 representing a perfect circle); and convex hull span ratio (the ratio of the major to minor axes of the convex hull, indicating the elongation of the enclosed shape).

### 4.8. Determination of Dopamine Levels

The striatal dopamine content was analyzed using ultra-performance liquid chromatography-tandem mass spectrometry (UPLC MS/MS). Freshly dissected striata were obtained under a Zeiss Stemi 2000-c stereoscopic binocular microscope (Zeiss, Göttingen, Germany) in ice-cold PBS and subsequently frozen in liquid nitrogen and maintained at −80 °C until analysis. Before overnight lyophilization, frozen samples were weighed. The lyophilized samples were then reconstituted in 200 µL of chilled methanol, containing 10 µM hydroxytyrosol as an internal standard, and sonicated for 1 min. Cellular extracts were centrifuged at 12,000 rpm for 1 min at 4 °C and stored at −20 °C until analysis. Samples were injected into the UPLC system (Acquity UPLC I-Class System, Waters Corp, Milford, MA, USA). Dopamine levels were determined using a 1.8 µm column (Waters Acquity UPLC ™ HSS T3 column 2.1 mm × 100 mm), followed by detection with a low-resolution triple quadrupole XEVO TQ-XS mass spectrometer detector (Waters Corp). A calibration curve from 20 µM to 0.256 nM was prepared using a 10 mM stock solution of dopamine (in methanol/10 µM hydroxytyrosol), and reference values were obtained by plotting the area correlation between the analyte and the internal standard against each concentration (nM). The final concentration of dopamine was expressed as ng/mg of the striatal tissue.

### 4.9. Determination of Gene Expression

Brains were isolated from mice sacrificed 2 and 7 days after the last MPTP injection. SNpc and striatum were obtained by microdissection, frozen in liquid nitrogen, and stored at −80 °C until further use. The processes of RNA extraction, reverse transcription, and qPCR followed established protocols [15]. cDNA was used to quantify the gene expression of *Bdnf*, *Th*, *Dat*, *TNF-α,* and *IL-1β* through real-time quantitative RT-PCR using iQ SYBR Green Supermix (BioRad) according to the manufacturer’s instructions. *Gapdh* served as the normalization reference, and we estimated fold change expression with the Delta-Delta Ct method. The primer sequences used for each gene are provided in Appendix A.

### 4.10. Primary Mesencephalic Cell Culture

Mesencephalic neuron–glia culturing was performed as described previously [98] with certain modifications. In brief, ventral mesencephalon tissue was extracted from mouse embryos (at 12 to 14 days of gestation) under a dissecting microscope, and it was promptly mechanically homogenized in Ca^2+^/Mg^2+^-free HBSS containing D-glucose and 0.2 mM ascorbic acid. Isolated cells were then plated at a density of 2 × 10^5^ cells/cm^2^ on glass coverslips pre-coated with poly-D-lysine (Sigma) in 0.1 M borate buffer (pH 8.4). After 20 min at 37 °C, adhered cells were transferred to a neurobasal medium [supplemented with B27, 15% fetal bovine serum, 2 mM L-glutamine, 0.2 mM ascorbic acid, 100 U/mL penicillin, and 100 µg/mL streptomycin (all from Thermo Fisher Scientific)]. Then, 24 h after plating, cells were transferred to a serum-free defined medium and cultured in a humidified chamber at 37 °C in a 5% CO_2_ atmosphere. Half of the medium was replaced with fresh medium after 3 days of culture. Following 7–8 days of culture, cells were stimulated with 5 μM MPP^+^ in the presence or absence of cortistatin (100 nM) for 48 h. Cell death was quantified by counting the number of TH-positive cells using immunofluorescence analysis as follows. Cell cultures were fixed with 2% PFA for 15 min at room temperature, blocked with 10% goat serum in PBS + 0.1% Triton X-100 for 1 h at room temperature, and subsequently incubated with anti-TH or anti-MAP2 antibodies (at 4 °C, overnight), followed by incubation with the corresponding secondary antibodies (Alexa Fluor 594 and 488) for 1 h at room temperature. Nuclei were counterstained with DAPI, and the samples were mounted with Mowiol. Extensive washing with PBS was performed between steps, and the samples were finally observed under a fluorescence microscope (Olympus IX81F-3; Olympus, Tokyo, Japan).

### 4.11. Statistical Analysis

All data are expressed as the mean ± standard error of the mean (s.e.m). The number of independent animals (*N*) and tissue sections, cellular types, and cell cultures (*n*) are shown. Statistical comparisons between two groups with parametric data and normal distribution were conducted using the unpaired Student’s *t*-test or the non-parametric Mann–Whitney *U* test. For three or more groups with a normal distribution and parametric data, a one-way ANOVA with appropriate post-hoc tests for multiple comparisons was employed (with a small number of data points, a Bonferroni post-hoc test was preferably used over the Tukey post-hoc test). The Kruskal–Wallis test, followed by Dunn’s post-hoc test, was performed with non-parametric data from three or more groups. In cases where standard deviations were assumed to be different, Brown–Forsythe and Welch ANOVA tests were applied with the Dunnett T3 post-hoc test. All analyses were conducted using the GraphPad Prism v8.3.0 software. *p* values < 0.05 (two-tailed) were considered significant. 

## Figures and Tables

**Figure 1 ijms-25-00694-f001:**
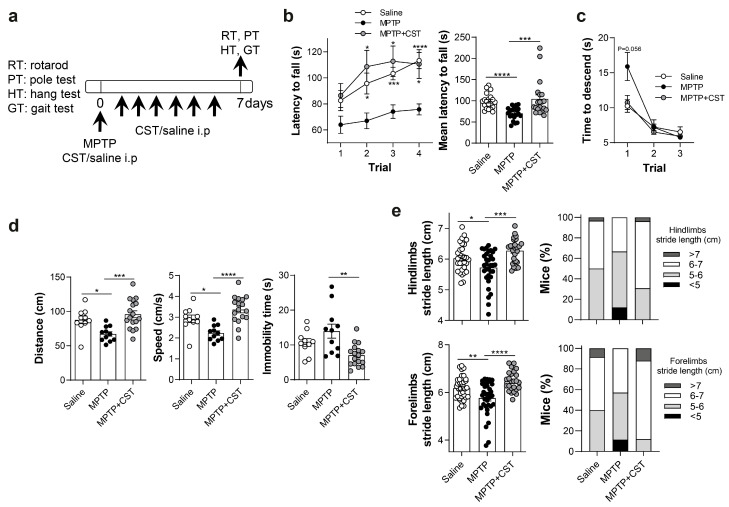
Treatment with cortistatin reduces deficits in MPTP-induced locomotor activity. (**a**) C57BL/6 mice were administered MPTP (20 mg/kg; 4 i.p. injections at 2 h intervals) and then treated intraperitoneally with vehicle (MPTP group) or cortistatin (MPTP + CST group) for seven consecutive days, beginning 2 h after the last MPTP injection. Animals injected intraperitoneally with saline instead of MPTP were used as controls (saline group). At day 7 post-MPTP infusion, deficits in movement coordination and locomotor activity were assessed in mice using several behavior tests, including the rotarod (RT), pole (PT), hang (HT), and gait (GT) tests. (**b**) Movement coordination was assessed on the rotarod. Data represent the latency in seconds to fall from a rotating rod accelerating from 4 to 40 rpm in 300 s across four consecutive trials (**left panel**) and the mean latency to fall for each mouse in the four trials (**right panel**); *N* = 20 mice/group from four independent experiments. (**c**) Locomotor activity was assessed using the pole test. The graph displays the time that mice needed to descend from the 45 cm pole after three independent trials; *N* = 12 (saline), 14 (MPTP), and 17 (MPTP + CST) mice from two independent experiments. (**d**) Neuromuscular strength was determined using the hang test. Mice were positioned on a horizontally inverted grid, and their locomotor activity (distance traveled and speed) and resting time were recorded during three trials; *N* = 10 (saline), 11 (MPTP), 17 (MPTP + CST) mice from two independent experiments. (**e**) Basal ganglia-related movement disorders were assessed with the gait test: stride lengths in the hindlimb (**top panels**) and forelimb (**bottom panels**) were measured as the distance between successive paw prints. The average of two strides was calculated for each animal (**left panels**); the percentage of mice in each group with short (<5), medium (5–6), long (6–7), and very long (>7) strides (all in cm) is represented (**right panels**). *N* = 25–35 mice/group from four independent experiments. All data are expressed as mean ± s.e.m. Each dot represents an individual mouse. * *p* < 0.05, ** *p* < 0.01, *** *p* < 0.001, **** *p* < 0.0001 vs. MPTP-treated mice.

**Figure 2 ijms-25-00694-f002:**
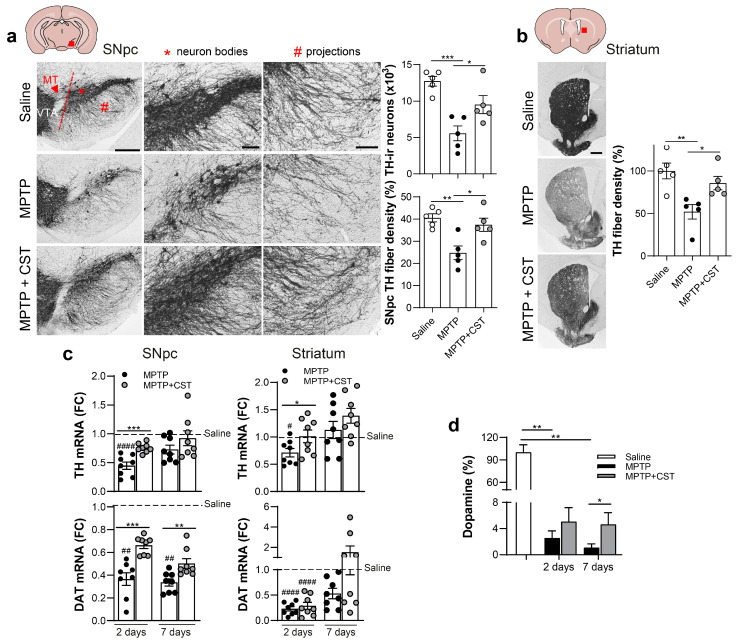
Neuroprotective effect of cortistatin on the dopaminergic nigrostriatal pathway affected by MPTP intoxication. Experimental PD was induced in mice and treated with cortistatin, as shown in Figure 1a. Brains were isolated two and seven days after MPTP injection. (**a**,**b**) Coronal sections corresponding to the SNpc-containing ventral mesencephalic (**a**) and striatum (**b**) regions of brains isolated on day 7 were immunostained for tyrosine hydroxylase (TH, depicted in black staining). Representative sections from each brain region (indicated by red squares in the schemes) are shown. Scale bars: 500 µm for general views in (**a**,**b**), 200 µm for detailed views of neuron bodies (pointed out by a red asterisk) and projections (pointed out by a red hash) in (**a**). SNpc: substantia nigra pars compacta; VTA: ventral tegmental area; MT: medial terminal nucleus. The number of TH-ir neurons in SNpc was quantified through stereology (**a**, **top right panel**). TH^+^ innervation (fiber density) was evaluated through densitometry analysis in both nigrostriatal regions (**a**,**b**, **right panels**). *n* = 5 sections/mouse; *N* = 5 mice/group. (**c**) Gene expression of *Th* and dopamine transporter (*Dat*) was determined using real-time qPCR in mRNA isolated from SNpc (**left panels**) and the striatum (**right panels**) of brains that were dissected at the indicated times post-MPTP. Gene expression was normalized to *Gapdh* expression, and data are shown as the fold change in the mRNA levels for each gene in brain tissues isolated from saline-treated mice (set at 1; dashed line). *N* = 8 mice/group. (**d**) The content of dopamine in the striatum was determined as described in the methods and expressed as a percentage of the levels found in saline-treated mice. *n* = 5 (saline, MPTP + CST), 10 (MPTP) samples. All data are expressed as mean ± s.e.m. Each dot represents an individual mouse. * *p* < 0.05, ** *p* < 0.01, *** *p* < 0.001, vs. MPTP-mice. # *p* < 0.05, ## *p* < 0.01, #### *p* < 0.0001, vs. saline-treated mice.

**Figure 3 ijms-25-00694-f003:**
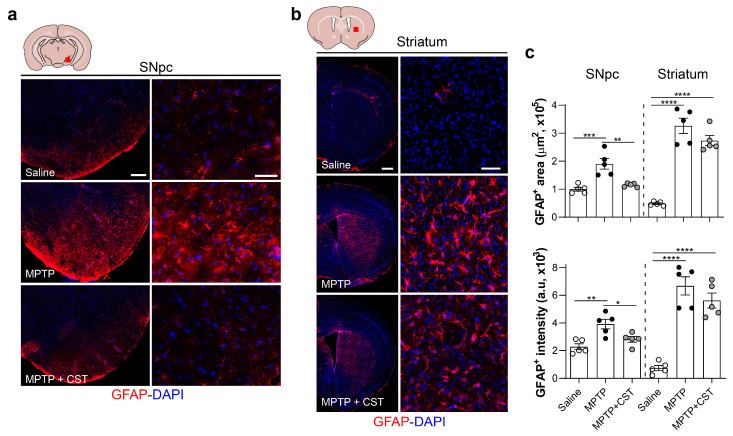
Cortistatin diminishes nigral astrocytic activation induced by MPTP intoxication. Experimental PD was induced in mice and treated with cortistatin, as shown in Figure 1a. Brains were isolated seven days after MPTP injection, and coronal sections corresponding to the SNpc-containing ventral mesencephalic (**a**) and striatum (**b**) regions were immunostained for GFAP (red staining). The localization of both regions is indicated with a red square in the upper schematic drawing. Images show representative sections that are consecutive to those used in Figure 2 for TH detection. Scale bars: 200 µm in (**a**) and 500 µm in (**b**) for general views (**left panels**) and 50 µm for detailed views (**right panels**). Nuclei were counterstained with DAPI (blue). (**c**). Quantification of astrogliosis in both brain regions was performed by measuring the GFAP^+^-occupied area (**top**) and the average GFAP intensity (**bottom**). Mice injected with saline were used as reference controls. *N* = 5 mice/group. All data are reported as mean ± s.e.m. Each dot represents an individual mouse. * *p* < 0.05, ** *p* < 0.01, *** *p* < 0.001, **** *p* < 0.0001.

**Figure 4 ijms-25-00694-f004:**
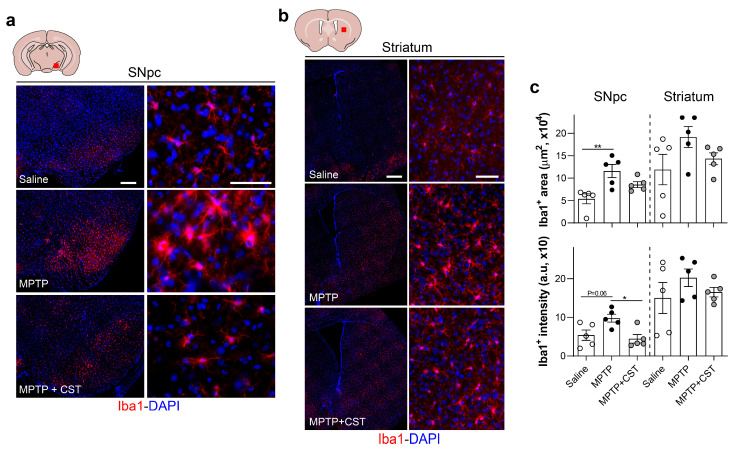
Treatment with cortistatin regulates MPTP-induced reactive microglia in the SNpc. Experimental PD was induced in mice and treated with cortistatin, as shown in Figure 1a. Brains were isolated seven days after MPTP injection, and coronal sections corresponding to the SNpc-containing ventral mesencephalic (**a**) and striatum (**b**) regions were immunostained for Iba1 (red staining). The localization of both regions is indicated with a red square in the upper schematic drawing. Images show representative sections that are consecutive to those used in Figure 2 for TH detection. Scale bars: 200 µm in (**a**) and 500 µm in (**b**) for general views (**left panels**) and 50 µm for detailed views (**right panels**). Nuclei were counterstained with DAPI (blue). (**c**). Quantification of microgliosis in both brain regions was performed by measuring the Iba1^+^-occupied area (**top**) and the average Iba1 intensity (**bottom**). Mice injected with saline were used as reference controls. *N* = 5 mice/group. All data are reported as mean ± s.e.m. Each dot represents an individual mouse. * *p* < 0.05, ** *p* < 0.01.

**Figure 5 ijms-25-00694-f005:**
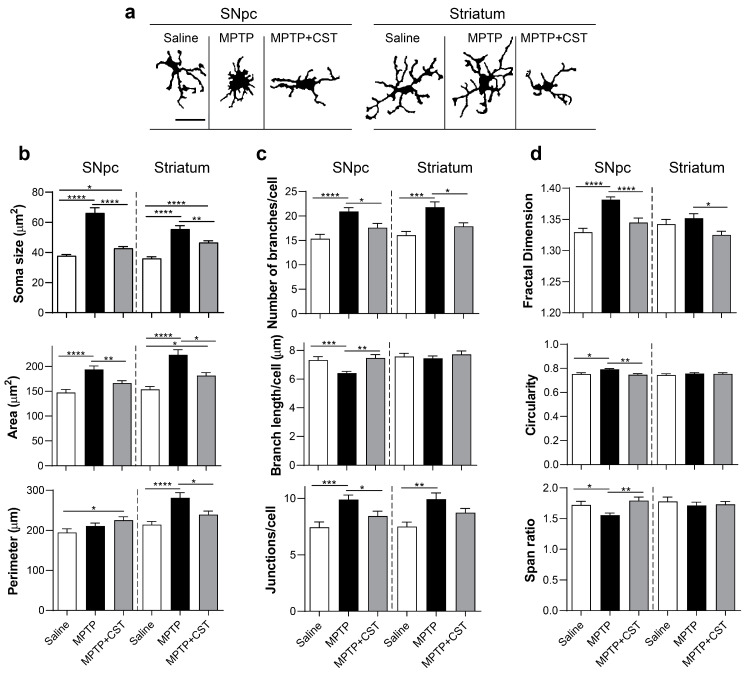
Cortistatin modulates microglia morphotypes after MPTP intoxication. Experimental PD was induced in mice and treated with cortistatin, as shown in Figure 1a. Coronal sections were obtained from the SNpc and striatal regions of brains dissected on day 7 after MPTP and immunostained for Iba1. Iba1^+^ microglial cells were morphometrically analyzed. (**a**) Representative binary images showing morphological prints of selected Iba1^+^ cells from SNpc and striatum obtained from saline, MPTP, and cortistatin-treated mice (MPTP + CST). Scale bar: 20 µm. (**b**–**d**) Microglia morphology descriptors based on the analysis of the cell shape ((**b**), soma size, area, and perimeter), cellular network of projections ((**c**), skeleton evaluation of the number and length of branches and the number of junctions), and cellular complexity ((**d**), fractal dimension, circularity, and span ratio). *n* = 65–75 cells/saline, *n* = 75–90 cells/MPTP, and *n* = 70–85 cells/MPTP + CST (for both SNpc and striatum) were analyzed; *N* = 5 mice/group. All data are reported as mean ± s.e.m. * *p* < 0.05, ** *p* < 0.01, *** *p* < 0.001, **** *p* < 0.0001.

**Figure 6 ijms-25-00694-f006:**
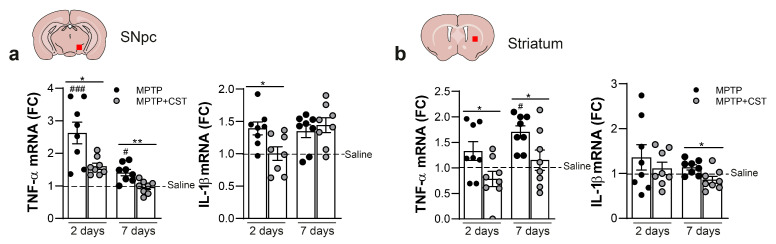
Cortistatin affects the expression profile of inflammatory factors in the nigrostriatal pathway of MPTP-intoxicated mice. C57BL/6 mice were administered saline or MPTP (20 mg/kg; 4 i.p. injections at 2 h intervals) and treated intraperitoneally with cortistatin (MPTP + CST) for seven consecutive days. The midbrains were isolated two or seven days post-MPTP and dissected as described in the Methods section. Gene expression levels in SNpc (**a**) and striatum (**b**) were quantified using real-time qPCR analyses and normalized to *Gapdh* expression. The data are represented as the fold change in the mRNA levels for each gene in each brain tissue isolated from saline-treated control mice (set at 1; dashed line). *N* = 8 mice/group. All data are reported as mean ± s.e.m, with dots representing individual samples. * *p* < 0.05, ** *p* < 0.01, vs. MPTP-mice; # *p* < 0.05, ### *p* < 0.001, vs. saline-treated mice.

**Figure 7 ijms-25-00694-f007:**
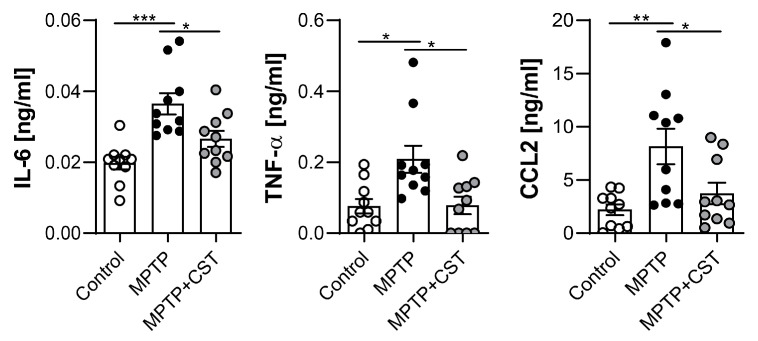
Cortistatin reduced the expression of systemic inflammatory factors. The levels of the proinflammatory cytokines IL-6 and TNF-α, as well as the chemokine CCL2, were determined using ELISA in the sera of saline, MPTP, and MPTP + CST-treated mice (isolated 48 h after the last saline or MPTP injection). *N* = 10 mice/group. All data are reported as means ± s.e.m, with dots representing individual mice. * *p* < 0.05, ** *p* < 0.01, *** *p* < 0.001, vs. MPTP-mice.

**Figure 8 ijms-25-00694-f008:**
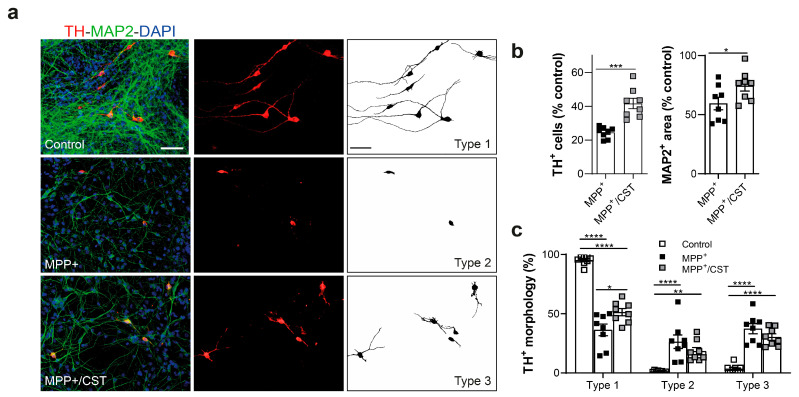
Cortistatin protects dopaminergic neurons against MPP^+^-induced cell death. Ventral midbrain cultures were derived from the mesencephalon of embryonic mice (E13.5-E14) and exposed to MPP^+^ (5 μM) for 48 h in the absence (MPP^+^) or presence of cortistatin (100 nM) (MPP^+^/CST). (**a**) Representative images depicting the density of dopaminergic neurons (TH^+^, red) relative to total neurons (identified with the microtubule-associated protein 2, MAP2^+^, green) in each condition. Nuclei were stained with DAPI (blue). Scale bar: 50 µm. (**b**) The number of dopaminergic (**left panel**) and non-dopaminergic (**right panel**) neurons in the cultures was determined by counting the TH^+^ and MAP2^+^ cells, respectively, and expressed as a percentage of the total TH^+^ (left) or MAP2^+^ (right) neurons found in control cultures (treated with medium alone). (**c**) Relative percentage of TH^+^ neurons based on distinct morphologies identified in the binary images obtained from panel (**a**) (Type 1: neurons with long dendrites; Type 2: neurons lacking dendrites with swollen cell bodies; Type 3: neurons with dendritic shortening). *N* = 8 independent cell cultures/group. All data are reported as mean ± s.e.m, with dots representing individual cultures. * *p* < 0.05, ** *p* < 0.01, *** *p* < 0.001, **** *p* < 0.0001.

## Data Availability

All relevant data were included in the paper and Appendix A. This study did not generate datasets deposited in external repositories.

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
