# Peer review of "Cortistatin as a Novel Multimodal Therapy for the Treatment of Parkinson’s Disease"

_ijms, 2024, doi:10.3390/ijms25020694_

Round 1
Reviewer 1 Report
Comments and Suggestions for Authors
The manuscript entitled “Cortistatin as a novel multi-modal therapy for the treatment of Parkinson´s Disease” by Serrano-Martínez et al. describes the beneficial effects of the neuropeptide Cortistatin in the MPTP animal model for Parkinson´s Disease (PD). The administration of Cortistatin prevented the loss of dopaminergic neurons in the substantia nigra and the striatum caused by MPTP injections, with important improvements in the clinical symptoms of MPTP-treated mice. The authors also observed a reduction of gliosis and of immune mediators by the treatment with Cortistatin, as well as a decrease of dopaminergic cell death in in vitro experiments.
Overall, the manuscript is well written and the experimental set up is very complete, supporting their hypothesis with a variety of data. However, the authors also use an inadequate nomenclature for microglia phenotypes and should adapt their wording to the new recommendations, published by nearly 100 authors of the microglia field in Paolicelli et al., 2022. In addition, the quality of the figures in the manuscript is very low. This might be due to the conversion to pdf in the manuscript format, since the quality of the figures in the Supplementary Material is perfect.
Major comments:
1. Throughout the text (including the Supplementary Material), the authors have to adopt their nomenclature to the recommendations given by Paolocelli et al., 2022. Especially the terms “activated microglia”, “resting microglia” and “microglial activation” must be avoided.
2. Figure 8: The authors claim that Cortistatin specifically prevents the cell death of TH+ neurons. However, from the pictures in Figure 8 it seems that the treatment with MPP+ alone and with MPP+/CST also affects the viability of all other neurons (see MAP2 staining under different conditions). The author should estimate also this loss/gain of total neurons (maybe by quantification of the total MAP2-positive area) and include this data in Figure 8. The text should be modified accordingly, including in the abstract. If Cortistatin at the end also affected the viability of other neurons (as it looks like from the images in this figure), it would not be so surprising, as its immunomodulatory effects have been observed to be beneficial in other neuropathologies. There might be also a general loss of neurons, as the glial cells present in the culture react to MPP+ and may secrete neurotoxic factors.
In this figure, the binary images in the right panel (Type 1, Type 2, Type3) only correspond in the MPP+ condition to the immunofluorescence picture. To avoid unnecessary confusion to the readers, it would be helpful to depict the binary images of the cells shown by the immunofluorescence.
3. Sometimes the authors tend to generalize their results, leading to overinterpretation of their data in some cases:
For example, BDNF expression levels are restored by Cortistatin only after 7 days in the striatum. Thus, the authors should specify this in the abstract (for example “…promoted the expression of neurotrophic factors in the striatum”.), remove “throughout the experiment” on page 6 line 209 (or replace this term by “at this time point”), and replace “preferentially” by “only” on page 13, line 405.
Similarly, only the intensity of Iba+ cells is reduced in the SNpc and not in the striatum. The authors should not generalize this effect, as it was done in the description for Figures 3, 4, S3 and S4 on page 6 lines 223 - 229 and on page 13, lines 445-446. In this sense, the title of Figure S4 should be adopted to a more neutral one.
In addition, Th mRNA expression was only affected significantly at early time points. Thus, the authors should remove “…and later time points” on page 5 line 181.
Minor comments:
1. As IL-6 can exert dual immunomodulatory and neurotrophic functions in the CNS, why did the authors only demonstrate results about IL-6 at the serum level and did not test the IL-6 expression level in the SNpc and the striatum, as they did with TNF-alpha and IL-1beta? Can the authors provide any data about IL-6 at the CNS-level?
2. Page 5, lines 181-182: As in the Figure 2 the data of TH-immunoreactive neurons come first, I would recommend to keep the same order in the text.
Comments on the Quality of English LanguagePage 2, line 104: “in” should be “of”
Page 4, line 166: “induce” should be “induced”
Page 6, line 224: Keep only the abbreviation SNpc.
Page8, line 284: I would recommend to replace “Conversely” by “Interestingly” or similar.
Page 9, line 288: I would recommend to remove “complex”, but it is up to the authors.
Page 17, line 617: “yhem” should be “them”
Page 17, line 618-619: Please revise the English of the sentence “As mice run away in the paper, paws kept marked.”. For example: “By walking on white paper, foot prints were recorded and analyzed.” or similar.
Supplementary Figure S5 legend: Remove “itself” and “analized” should be “analyzed”.
Reviewer 2 Report
Comments and Suggestions for Authors
In this study, the authors probed the putative effects of cortistatin, a small molecule bioactive peptide, in improving symptoms and neurodegenerative processes in rats treated with 1-methil-4-phenyl1-1,2,3,6-tetrahydropyridine (MPTP), as an animal model of Parkinson’s disease (PD). The authors demonstrated that rats treated with cortistatin presented reduced dopaminergic loss when compared with MPTP rats. Also, MPTP rats under cortistatin experienced a significative improvement of gait abnormalities.
This study would be of potential interest for a wide population of researchers since it focuses on a novel therapeutic strategy in PD: the neuroprotection of neuroinflammation beyond current symptomatic treatment in PD, including L-Dopa. Therefore, the authors stated that cortistatin would be a possible disease-modifying drug in PD and this would be a scientific breakthrough. However, I have several major concerns about the rationale of study.
First, although there are reports on the possible role of neuroinflammation in the pathophysiology of dopaminergic loss in PD, the most credited pathophysiologic hypothesis pointed on the neurodegenerative process and its cascade. Therefore, the inflammation could be an iceberg peak, an epiphenomenon of a far more complex process. How do the authors account for this?
The experimental model of MPTP PD can apply when considering PD-related signs and symptoms but it is far away from matching the complex pathophysiology of neurodegeneration in PD.
Extensive literature in the field of the role of cortistatin in neurodegenerative disorders is missing.
Comments on the Quality of English LanguageExtensive editing of English language required
Round 2
Reviewer 1 Report
Comments and Suggestions for Authors
The authors have addressed all my concerns and the manuscript is now suitable for publication in the International Journal of Molecular Sciences. Though, the authors still need to correct some errors in the English language. Some examples are given below. The authors also use the word “to show” quite often. In some occasions they could replace it by “to demonstrate”.
In addition, the authors should avoid using the term “phagocytic morphology” as microglia can phagocyte in a variety of different cell shapes.
Comments on the Quality of English LanguagePage 1 line 41: “reduction in the cell death” should be a “reduction of cell death”
Page 1 line 49: “Along last generation” should be rephrased, for example “during the last decades” or similar.
Page 2 line 67: “PD patients have showed…” should be a “PD patients have shown…”
Page 2 line 75: “…have showed…” should be a “…have shown…”
Page 4 line 159: “…on the grid for longer periods of time than that showed by MPTP-treated mice” should be “…longer than MPTP-treated mice on the grid”.
Page 5 line 182: “…after MPTP” should be after “MPTP treatment”.
Page 6 line 211: “…similar Bdnf levels than those found” should be “… similar Bdnf levels as in saline-injected….”
Page 6 line 229: “Heightened” should be “elevated” or similar.
Page 8 line 277: “higher occupied area” should be “increased area...”.
Page 8 line 278: “more number of branches” should be “more branches”.
Page 8 line 280: “small cell bodies, mild ramified cells with long thin processes arising from the cytosol” should be “small cell bodies with long thin processes arising from the cytosol, representing mildly ramified cells.” or similar.
Page 9 line 287: “that observed in samples” should be “the ones in control mice”.
Page 9 line 289: “than that observed in microglia of MPTP-treated mice” should be “than the ones of microglia from MPTP-treated mice”.
Page 10 line 307: “was sustained one week later” should be “was sustained up to 7 days”.
Page 12 line 363: “impacts” should be “affects”.
Page 12 line 382: “it could be highly promising searching” should be “it could be highly promising to search for…”
Page 13 line 445: “response” should be “respond” or “reply”.
Page 14 line 469: “…it resembles more of a phagocytic morphology” should be “… it rather resembles a phagocytic morphology”
Page 14 line 476: “… morphology similar to than observed” should be “morphology similar to the one observed.
Page 15 line 515: Please rephrase this part of the sentence “Considering that cortistatin has demonstrated the ability to reduce the production of inflammatory cytokines and nitric oxide by reactive glial cells.”
Page 15 line 540: “than that showed by wild-type mice when exposed to MPTP” shoud be “than wild-type mice, when exposed to MPTP.”
Page 16 line 569: “in” should be “from”.
Reviewer 2 Report
Comments and Suggestions for Authors
No further concerns
Comments on the Quality of English Language
English editing is required thorough the manuscript
